# Improved gene regulatory network inference from single cell data with dropout augmentation

Hao Zhu *, Donna K. Slonim *

Department of Computer Science, Tufts University, Medford, Massachusetts, United States of America

* donna.slonim@tufts.edu (DKS); hao.zhu@tufts.edu (HZ)

**Data availability statement:** The source code for this project (software + preprocessing scripts) is available at https://github.com/ TuftsBCB/dazzle. The processed data used in

## Abstract

A major challenge in working with single-cell RNA sequencing data is the prevalence of "dropout," when some transcripts' expression values are erroneously not captured. Addressing this issue, which produces zero-inflated count data, is crucial for many downstream data analyses including the inference of gene regulatory networks (GRNs). In this paper, we introduce two novel contributions. First, we propose Dropout Augmentation (DA), a simple but effective model regularization method to improve resilience to zero inflation in single-cell data by augmenting the data with synthetic dropout events. DA offers a new perspective to solve the "dropout" problem beyond imputation. Second, we present DAZZLE, a stabilized and robust version of the autoencoder-based structure equation model for GRN inference using the DA concept. Benchmark experiments illustrate the improved performance and increased stability of the proposed DAZZLE model over existing approaches. The practical application of the DAZZLE model on a longitudinal mouse microglia dataset containing over 15,000 genes illustrates its ability to handle real-world single cell data with minimal gene filtration. The improved robustness and stability of DAZZLE make it a practical and valuable addition to the toolkit for GRN inference from single-cell data. Finally, we propose that Dropout Augmentation may have wider applications beyond the GRN-inference problem. Project website: https://bcb.cs.tufts.edu/DAZZLE.

## Author summary

The prevalence of false zeros in single-cell data, or "dropout," affects many downstream analyses. A common approach is to eliminate these zeros through data imputation. We propose an alternative solution that focuses on regularizing the model and increasing model robustness against dropout noise. Counter-intuitively, this is done by augmenting the input data with a small number of zeros to simulate additional dropout noise.

this study is deposited at https://zenodo.org/records/15762315. The BEELINE benchmark data was obtained from the authors of DeepSEM but could be regenerated using the provided data and code from BEELINE (https://github.com/Murali-group/Beeline). The actual gene expression data files used in BEELINE came from from GEO datasets with the following accession numbers: GSE81252 (hHEP), GSE75748 (hESC), GSE98664 (mESC), GSE48968 (mDC), and GSE81682 (mHSC). The Hammond microglia data is obtained from NCBI's GEO database under accession number GSE121654 (https://www.ncbi.nlm.nih.gov/geo/query/acc.cgi?acc=GSE121654).

**Funding:** The author(s) received no specific funding for this work.

**Competing interests:** The authors have declared that no competing interests exist.

Validation is performed on the task of gene regulatory network inference. Our proposed model, DAZZLE, which uses the dropout augmentation idea, shows improved performance and robustness.

## 1. Introduction

Gene Regulatory Network (GRN) inference from expression data offers a contextual model of the interactions between genes *in vivo*. [1–3]. Understanding these interactions is crucial to gain insight into the development, pathology, and key points of regulation that may be amenable to therapeutic intervention [4].

While GRN inference from bulk transcriptomic data has a long history, many recent studies consider the contextual specificity offered by single-cell RNA sequence data (scRNA-seq) [5]. Single cell RNA sequencing allows researchers to analyze transcriptomic profiles of individual cells, providing a more detailed and accurate view of cellular diversity than traditional bulk methods. However, opportunities come with challenges. A recent benchmark paper on GRN inference summarized the major issues in single-cell data that cause challenges for GRN inference: cellular diversity, inter-cell variation in sequencing depth, cell-cycle problems, and sparsity due to dropout [6].

Despite these challenges, many methods have been proposed for context-specific GRN inference from single-cell RNA-sequencing data alone. Among established methods, GENIE3 [7] and GRNBoost2 [8] are tree-based approaches, initially proposed for bulk data, that have been found to work well on single-cell data without modification. LEAP [9] estimates pseudotime to infer gene co-expression over several lagged windows, suggesting that the lags can be used to infer regulatory relationships. SCODE [10] and SINGE [11] apply a similar pseudotime idea, combined with ordinary differential equations (ODEs) and Granger causality ensembles, to model the results. PIDC uses partial information decomposition to incorporate mutual information among sets of genes, modeling cellular heterogeneity[12].

Other methods infer GRNs by integrating transcriptomic and other data sources. For example, SCENIC [13] starts by identifying gene co-expression modules using GENIE3/GRNBoost2, followed by identifying key transcription factors (TFs) that regulate these modules or regulons. scMTNI [14] studies GRNs in different cell clusters using a multi-task learning framework. GRNUlar [15] uses recently developed unrolled algorithms to infer undirected GRNs from single-cell data by incorporating TF information. NetREX-CF [16] performs optimizations based on prior GRN networks and uses collaborative filtering to address the incompleteness of prior data. PANDA [17] further optimizes prior GRN networks using massage passing. However, single cell data alone is much more widely available and accessible in specific contexts than integrated multi-omic data sets.

The application of neural networks (NNs) in the analysis of single-cell data has advanced rapidly in the last couple of years. One of the leading NN-based GRN inference methods, DeepSEM, [18] parameterizes the adjacency matrix and uses a variational autoencoder (VAE) [19] architecture optimized on reconstruction error. In fact, on the BEELINE benchmarks [6] where the "right" networks are (approximately) known, DeepSEM reports better performance than other methods and runs significantly faster than most.

However, as shown later in this paper, one of the issues with DeepSEM is that as training continues, the quality of the inferred networks may degrade quickly. A possible explanation is that soon after the model converges, it may begin to over-fit the dropout noise in the data.

Single-cell data is often characterized by an excessive number of zero expression counts, referred to as "zero-inflation." For example, in the nine data sets examined in [20], 57 to 92

percent of the observed counts are zeros. Among these zero values, "dropout" describes the situation when transcripts, often those with low or moderate expression in a cell, are not counted by the sequencing technology. Later droplet-based protocols, such as inDrops [21] and 10X Genomics Chromium [22], helped improved detection rates. However, the "dropout" problem still persists, as even recent methods have relatively low sensitivity [23,24].

Therefore, there has been research into data imputation methods for use in single-cell analysis. Several methods have been proposed to identify and replace missing data with imputed values [23–27]. Yet many of these methods depend on restrictive assumptions, and some require additional information, such as GRNs or bulk transcriptomic data.

In this paper, we introduce two novel contributions to the fields of single cell analysis and GRN inference. First, we propose "dropout augmentation" (DA), a novel approach to mitigate the impact of the zero-inflation problem by augmenting the data with a small amount of simulated dropout noise. We found that this idea, while seemingly counter-intuitive, can effectively regularize models so that they remain robust against dropout noise.

It has long been known that by adding noise to the input data during training, we can improve the robustness, and sometimes even the performance, of many machine learning models. Bishop first pointed out that adding noise is equivalent to Tikhonov regularization [28]. Hinton further introduced the idea of using random "dropout" on either input or model parameters to improve training performance [29]. Thus, the theoretical foundations of DA are also solid.

Our second contribution is the DAZZLE model, or *Dropout Augmentation for Zero-inflated Learning Enhancement*. DAZZLE uses the same VAE-based GRN learning framework introduced by DeepSEM and DAG-GNN [18,30], but it employs dropout augmentation and several other model modifications. These include a new method optimizing the adjacency matrix sparsity control strategy, a simplified model structure, and a closed-formed prior. Compared to DeepSEM, DAZZLE shows better model stability and robustness in our benchmark experiments. We further illustrate how DAZZLE's network inference facilitates interpreting typical-sized data sets efficiently, in this case explaining microglial expression dynamics across the mouse lifespan. (This noise augmentation concept has been further developed in our RegDiffusion software [31], which relies instead on a diffusion-based learning framework.)

## 2. Results

### 2.1. Dropout augmentation and the DAZZLE model

GRN inference in DAZZLE is based on the structure equation model (SEM) framework previously employed by DAG-GNN and DeepSEM [18,30]. The input of the model is the gene expression matrix representing the scRNAseq data, where each raw count of $x$ is transformed to $log(x + 1)$ to reduce the variance and avoid taking the log of zero. We assume that the rows of the input matrix represent cells and the columns represent genes. The adjacency matrix $A$ is parameterized and used in both sides of an autoencoder, as shown in Fig 1. The model input is simply a single-cell gene expression matrix, where the rows correspond to cells and the columns to genes. The model is trained to reconstruct the input while the weights of the trained adjacency matrix are retrieved as a by-product of training. Since ground truth networks are never available to the training model, this type of SEM model should be considered an unsupervised learning method for GRN inference. We include a detailed explanation of why the learned adjacency matrix $A$ represents the underlying GRN, as well as other methodological details for DA and DAZZLE, in the Methods section.

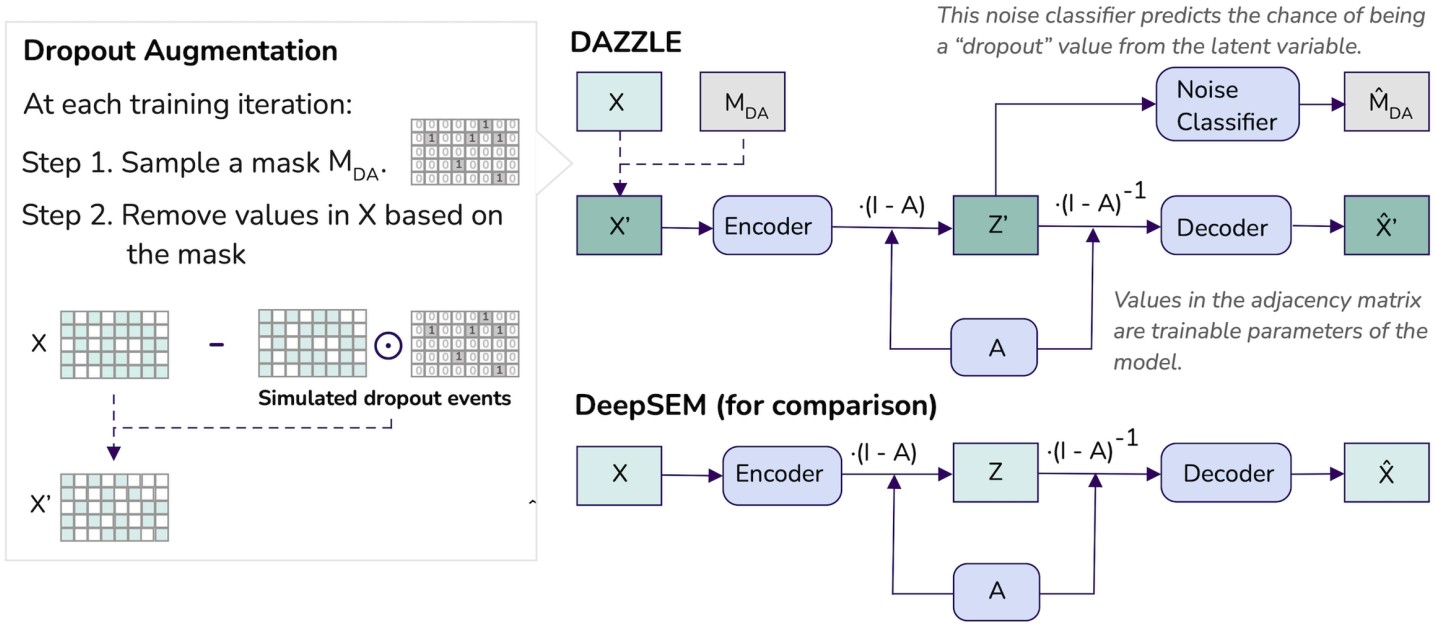

**Fig 1. One of the major differences between DAZZLE and DeepSEM is the use of Dropout Augmentation.** Dropout augmentation regularizes model training by simulating small amounts of random dropout at each training iteration such that the model is protected against the negative impact of dropout noise. Rounded boxes indicate trainable model parameters.

One unique design aspect that differentiates DAZZLE from DeepSEM is the use of DA, as shown in Fig 1. As a model regularization method, DA can be applied to any model design that is continuously optimized. At each training iteration, we introduce a small amount of simulated dropout noise by sampling a proportion of the expression values and setting them to zeros. With multiple training iterations, the model is exposed to multiple versions of the same data with slightly different batches of dropout noise. As a result, it is less likely to over-fit any particular batch.

DAZZLE also includes a noise classifier to predict the chance that each zero is an augmented dropout value; this classifier is trained together with the autoencoder. Since we generate the locations of the augmented dropout, we can confidently use them for training. The purpose of this classifier is to move the values that are more likely to be dropout noise to a similar region in the latent space $Z'$, so that the decoder will learn to put less weight on them when reconstructing the input data.

We made several additional model design and training choices that further distinguish our model from that of DeepSEM. First, we improved the stability of the model by delaying the introduction of the sparse loss term by a customizeable number of epochs. Another difference is that, to estimate the prior, DeepSEM estimates a separate latent variable while DAZZLE uses a closed-form Normal distribution. These changes lead to reduced model sizes and computational time. For example, to process the BEELINE-hESC dataset with 1,410 genes, the original DeepSEM implementation used 2,584,205 parameters and ran in 49.6 seconds (clock time) on an H100 GPU. Eliminating some unnecessary calculations, our DAZZLE implementation reduces the model to 2,022,030 parameters (a 21.7% reduction) without changing the size of the hidden layers. On the same device, our implementation finished inference in 24.4 seconds (a 50.8% reduction in running time). Finally, while DeepSEM is trained with two separate optimizers in an alternating manner (one on the adjacency matrix and the other on

the rest of the neural networks), DAZZLE is trained using a single optimizer with different learning rates. This improvement helps DAZZLE stay modular, so it can be integrated with other network components more easily in the future.

## 2.2. DAZZLE improves GRN inference on BEELINE benchmarks

We performed a benchmark comparison of DAZZLE to DeepSEM [18], GENIE3 [7], GRN-Boost2 [8], and PIDC [12] on the BEELINE single cell benchmark, which includes seven datasets (two from human and five from mouse) and three sets of ground truth networks. The details of the BEELINE benchmarks are described in the Methods section. We chose to compare to GENIE3 and GRNBoost2 as representatives of non-deep-learning methods because these two decision-tree-based methods are among the most widely used GRN inference algorithms. PIDC is another strong baseline method using mutual information on triplets of genes. Recent benchmarks on single cell data [32], [6] also showed that these methods generally outperformed SCODE [10], ppcor [33], and SINCERITIES [34]. We also included DeepSEM, the model that inspired this work and the previous front-runner.

Note that a single run of DeepSEM is not stable (a point that we discuss further in later Results sections). Thus, DeepSEM was proposed as an ensemble algorithm, combining results from a set of 10 runs. Here, we include both single runs (1x) and the ensemble model (10x) in the comparison for both DAZZLE and DeepSEM. For fair comparison, both DAZZLE and DeepSEM use the same size neural networks, and both are trained for 120 iterations as suggested by the DeepSEM paper. The hyperparameter settings are identical except for the changes already mentioned in Sect 2.1 (see also the comparison of hyperparameters in Sect 2 of S1 Text).

The main benchmark results are provided in Table 1. The numbers reported are the average area under the precision recall curve (AUPRC) ratios over 10 runs, where higher values represent better performance. In Table 1, the evaluation is done separately for the STRING network, the Non-celltype-specific ChIP-Seq network, and the Celltype-specific ChIP-Seq network. Note that for celltype-specific ChIP-Seq, we followed the recommendations of the DeepSEM paper and applied a very small L1 sparsity regulation to the adjacency matrix for better performance.

Among all the evaluations in Table 1 the ensemble version of DAZZLE (DAZZLE-10x) has the best performance of all methods in over half the cases, and in all the other cases it ranks either 2nd or 3rd, with a result that is within 6% of the top score. The DAZZLE-10x results also typically have lower variance compared to the DeepSEM-10x results, illustrating the stability of the DAZZLE model. Further, across nearly all benchmarks, a single run of DAZZLE, DAZZLE-1x, has a higher AUPRC ratio than the comparable single-run DeepSEM-1x. Yet, DAZZLE-1x is typically also more stable. Some DeepSEM-1x results in Table 1 have substantial variance, but this unwanted model behavior is eliminated in DAZZLE-1x; the reasons for this are explained in Sect 2.4. We see similar findings when we use Early Precision Ratio (EPR) or the Area Under the Receiver Operating Curve (AUROC) as the evaluation metrics (Supplement Tables B and C in S1 Text).

For the cell-type specific data, the highest observed AUPRC ratio for any data set or method is 1.10, and the highest observed EPR is 1.20 (just slightly better than random performance), suggesting that for all methods, reproducing these cell-specific networks is nearly impossible. A possible explanation of this effect appears in S2 Fig, which shows that the results are highly variable across training iterations. We confirmed that using very low L1 regularization yields better performance on this ground truth network; the reason for this is still unclear.

**Table 1. DAZZLE shows improved GRN inference capacity on BEELINE benchmarks.**

| | hESC | hHep | mDC | mESC | mHSC-E | mHSC-GM | mHSC-L |
|---|---|---|---|---|---|---|---|
| # of Genes | 1410 | 1448 | 1321 | 1620 | 1204 | 1132 | 692 |
| # of Cells | 758 | 425 | 383 | 421 | 1071 | 889 | 847 |
| **Ground Truth: STRING** | | | | | | | |
| # of True Edges | 5,149 | 9,000 | 5,898 | 8,479 | 1,826 | 1,311 | 154 |
| GENIE3 | 1.98 (0.01) | 1.86 (0.01) | *1.72 (0.01)* | 2.05 (0.01) | 4.19 (0.05) | 6.27 (0.04) | 7.02 (0.08) |
| GRNBoost2 | 1.67 (0.01) | 1.50 (0.01) | 1.43 (0.01) | 1.88 (0.02) | 3.65 (0.05) | 5.16 (0.11) | 7.07 (0.08) |
| PIDC | 2.01 (0.00) | *1.90 (0.00)* | 1.58 (0.00) | 2.06 (0.00) | 5.09 (0.00) | 6.27 (0.00) | 6.72 (0.00) |
| DeepSEM-1x | 2.00 (0.06) | 1.66 (0.04) | 1.55 (0.06) | 2.20 (0.02) | 5.27 (0.16) | 5.93 (0.38) | 7.28 (0.60) |
| DeepSEM-10x | 2.10 (0.03) | 1.82 (0.05) | 1.68 (0.02) | 2.33 (0.03) | 5.68 (0.11) | *6.91 (0.12)* | 7.47 (0.14) |
| DAZZLE-1x | 2.39 (0.05) | 1.80 (0.02) | 1.58 (0.03) | 2.27 (0.04) | 5.87 (0.21) | 6.56 (0.10) | 7.50 (0.07) |
| DAZZLE-10x | *2.44 (0.02)* | 1.82 (0.01) | 1.62 (0.01) | *2.34 (0.03)* | 6.07 (0.03) | 6.63 (0.04) | *7.53 (0.05)* |
| **Ground Truth: Non-celltype-specific ChIP-Seq** | | | | | | | |
| # of True Edges | 4,597 | 5,335 | 3,918 | 8,030 | 1,960 | 1,358 | 317 |
| GENIE3 | 0.97 (0.00) | 1.01 (0.01) | 1.56 (0.01) | 1.65 (0.01) | 2.54 (0.02) | *3.56 (0.03)* | 2.78 (0.04) |
| GRNBoost2 | 1.01 (0.01) | 1.09 (0.01) | 1.36 (0.02) | 1.49 (0.02) | 2.39 (0.04) | 2.84 (0.03) | 2.45 (0.05) |
| PIDC | 1.13 (0.00) | 1.20 (0.00) | 1.65 (0.00) | 1.42 (0.00) | 2.65 (0.00) | 3.35 (0.00) | 2.91 (0.00) |
| DeepSEM-1x | 1.22 (0.05) | 1.29 (0.08) | 1.73 (0.07) | 1.58 (0.05) | 2.94 (0.32) | 3.27 (0.15) | 2.50 (0.56) |
| DeepSEM-10x | 1.24 (0.01) | 1.41 (0.03) | *1.93 (0.02)* | *1.66 (0.02)* | 3.18 (0.06) | 3.48 (0.12) | 3.01 (0.20) |
| DAZZLE-1x | 1.27 (0.05) | *1.44 (0.04)* | 1.85 (0.04) | 1.62 (0.05) | 3.30 (0.05) | 3.45 (0.06) | 3.00 (0.12) |
| DAZZLE-10x | *1.29 (0.01)* | 1.44 (0.01) | 1.89 (0.02) | 1.64 (0.01) | *3.35 (0.02)* | 3.51 (0.03) | *3.21 (0.11)* |
| **Ground Truth: Celltype-specific ChIP-Seq** | | | | | | | |
| # of True Edges | 7,050 | 15,410 | 1,193 | 42,795 | 21,975 | 14,135 | 5,180 |
| GENIE3 | 0.96 (0.00) | *1.10 (0.00)* | 1.01 (0.01) | 1.06 (0.00) | 0.99 (0.00) | 0.98 (0.00) | 1.05 (0.00) |
| GRNBoost2 | 1.01 (0.00) | 1.02 (0.00) | 1.00 (0.01) | 1.02 (0.00) | 0.99 (0.00) | 0.99 (0.00) | 1.01 (0.00) |
| PIDC | 0.98 (0.00) | 1.03 (0.00) | 1.04 (0.00) | 1.02 (0.00) | 0.95 (0.00) | 0.96 (0.00) | 0.99 (0.00) |
| DeepSEM-1x | 1.06 (0.02) | 1.03 (0.01) | 1.04 (0.03) | 1.03 (0.01) | 1.01 (0.00) | 1.02 (0.01) | 1.06 (0.01) |
| DeepSEM-10x | 1.07 (0.01) | 1.03 (0.00) | *1.05 (0.01)* | 1.03 (0.00) | 1.01 (0.00) | 1.02 (0.00) | 1.06 (0.00) |
| DAZZLE-1x | 1.10 (0.02) | 1.02 (0.01) | 1.03 (0.03) | 1.03 (0.01) | 1.01 (0.01) | 1.01 (0.01) | 1.08 (0.01) |
| DAZZLE-10x | *1.10 (0.01)* | 1.03 (0.00) | 1.05 (0.01) | *1.03 (0.01)* | *1.01 (0.00)* | 1.08 (0.00) | *1.08 (0.00)* |

Metric: AUPRC Ratio; Number of target genes: 1000.

Numbers reported are mean and std. of AUPRC Ratio compared with random guess over 10 runs. Higher ratios indicate better performance. Here, italicized text in dark shading indicate the best algorithms and lightly shaded cells indicate the 2nd best algorithms. In this experiment, we ran each algorithm on all available data with the default settings. The 10x models ensemble inferred networks from 10 runs and the 1x models are simply single-run models.

Further explanations of the model improvements are discussed in Sects 2.3 and 2.4.

In summary, under similar training conditions, DAZZLE outperforms DeepSEM and demonstrates more stable performance. We found that while a single run of DeepSEM can produce unstable results, the ensemble strategy proposed in the DeepSEM paper significantly enhances its performance and reduces variance. However, this improvement comes at the cost of a tenfold increase in execution time. For DAZZLE, the ensemble strategy also yields excellent results, making DAZZLE-10x the best performer in our comparison. Note however that the performance of DAZZLE-1x itself approaches that of an ensemble method, in essence because it functions as an ensemble method, as described in the Methods section below. For datasets with a large number of genes and cells, a single pass of DeepSEM-1x or DAZZLE-1x can take several hours to compute, even on modern GPUs. In such cases, a single pass of DAZZLE-1x offers a sufficiently-good solution at a more reasonable cost.

## 2.3. Dropout augmentation contributes to model robustness

In this section, we discuss the effects of dropout augmentation alone on GRN inference. S2 Fig shows the quality of the inferred networks from DeepSEM-1x as a function of the

number of training steps. Specifically, it shows that the quality of the inferred networks from DeepSEM tends to drop quickly if the model is over-trained, suggesting that the model may be overfitting to unwanted patterns in the training data. One possible solution is to stop training early, as suggested in the DeepSEM paper, which ends model training at 120 iterations. The tricky part is that the point of peak performance is very difficult to predict and may depend highly on the number of genes and cells in the dataset. In practice, when ground-truth networks are not available, it is very difficult to identify a good convergence point for the model. It would be ideal if inferred GRN accuracy remained robust, or at least if it dropped slowly and consistently after the performance peak, so that the cost of picking a sub-optimal stopping point would be low. We believe that dropout augmentation is potentially an effective solution to this problem.

For fairness, all the comparisons shown in Fig 2 were performed on his DAZZLE-1x. We varied only the percentage of augmented dropout values while keeping all the other hyper-parameters the same. Without DA (thick black lines, representing a DA probability of 0%), a common pattern is that the AUPRC Ratios drop after the performance peaks, similar to what we observed with the DeepSEM method (S2 Fig). However, when we train the model with

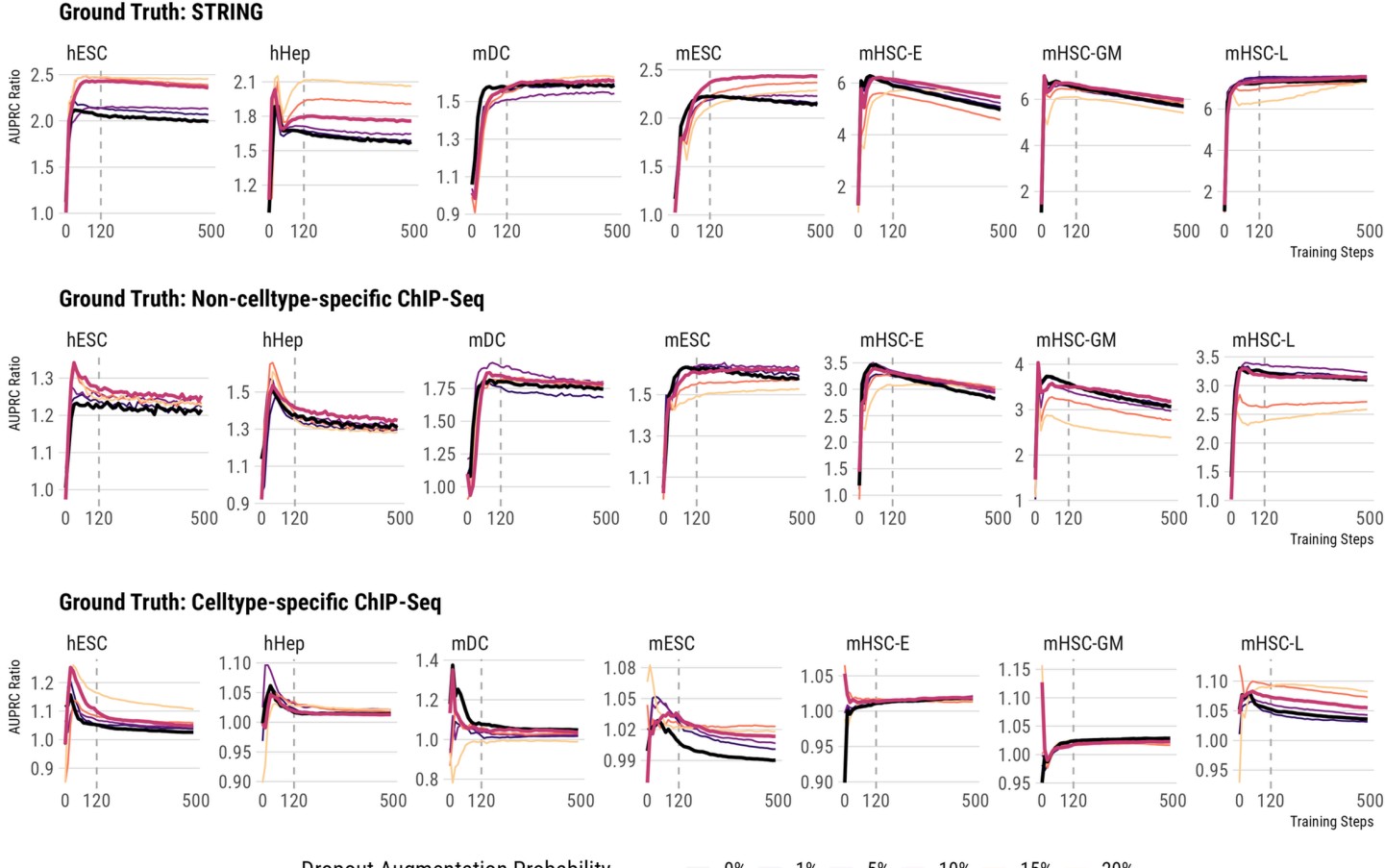

**Fig 2. Appropriate amount of augmented dropout helps maintain model robustness and may contribute to better performance.** Color reflects the probability of dropout augmentation. The two thick lines represent two important conditions, 0% - no augmented dropout, and 10% - the default dropout augmentation level we recommend. Dashed lines show the default number of training iterations used in DeepSEM and DAZZLE.

some amount of DA, in most cases the AUPRC Ratios either stay flat or decrease at a slower pace. For some data sets, such as hESC and hHep, DA also improves accuracy significantly. However, we observed that different datasets and ground truth networks seem to require different optimal levels of dropout augmentation. For hESC and hHep on the STRING network, 20% dropout augmentation yields the best AUPRC scores, but that amount of DA is too high for mHSC-E, mHSC-GM, and mHSC-L since it either slows down the model's convergence or leads to lower accuracy. After reviewing all cases, we recommend using 10% DA probability as the default for DAZZLE.

To further study the actual impact of DA, we conducted a controlled ablation study on the BEELINE benchmarks (with STRING as ground truth) by training variants of DAZZLE-1x (with different DA probabilities) and DeepSEM-1x under identical hyperparameter settings (NN learning rate = $1 \times 10^{-4}$, Adj matrix learning rate = $2 \times 10^{-5}$, batch size = 64). In all cases, we zeroed out a certain proportion of data points before the models saw the data to simulate background dropout noise. As shown in S1 Fig, DAZZLE-1x maintains its performance advantages over DeepSEM-1x in nearly all cases. In many cases, such as hESC, hHep, mDC, mHSC-E, mHSC-GM, and mHSC-L, DAZZLE-1x running on data with 10% additional background dropout noise performs better than DeepSEM-1x running on the full data set, demonstrating DAZZLE's resilience with zero-inflated data.

## 2.4. Sparsity control strategy improves DAZZLE's stability

In addition to increased robustness, DAZZLE also generates more stable GRN inference results thanks to its improved sparsity control strategy. As mentioned in Sect 2.2, results from DeepSEM-1x are not stable. Fig 3 examines this issue using 100 runs of DeepSEM-1x and DAZZLE-1x on the hESC dataset evaluated using the STRING network. In Fig 3a, we show histograms of the AUPRC ratios for both methods. The accuracy of the 100 runs of DeepSEM-1x can be separated into two groups. As shown in S3 Fig, other benchmark data sets produced similar results. Further investigation suggested that the L1 regularization on the adjacency matrix might be the main cause of those less ideal results.

Both DAZZLE and DeepSEM are trained with L1 regularization on the adjacency matrix. This regularization term ensures that the model doesn't add unnecessary weights to the adjacency matrix. It also ensures the singularity of the inferred adjacency matrix. Experiments have shown that a large enough coefficient for this sparse loss is required to generate a meaningful adjacency matrix prediction in GRN inference. However, as shown in Fig 3b, for DeepSEM-1x, where this L1 sparse loss is introduced at the very beginning of the training process, the optimization of this loss may be prioritized such that the values in the adjacency matrix quickly drop to near zero within the first several training steps. Most runs that do this end up in the low performance group. In DAZZLE, we propose a simple solution to overcome this limitation by delaying the introduction of the L1 sparse loss by a few training steps. As shown in Fig 3c, after 5 steps of training without the L1 sparse loss, the values and the gradients on the adjacency matrix parameters are stabilized sufficiently so the introduction of the sparse loss is less likely to destabilize the training process. We also see similar findings on other benchmark datasets; detailed results are included in S3 Fig.

## 2.5. Case study: Using DAZZLE to predict temporal changes in GRNs in mouse microglia

To test the effectiveness of running DAZZLE on more typical-sized sets of single-cell data, we applied the method to published data characterizing mouse microglia at different developmental stages [35]. For each time point, DAZZLE inferred an adjacency matrix for all the

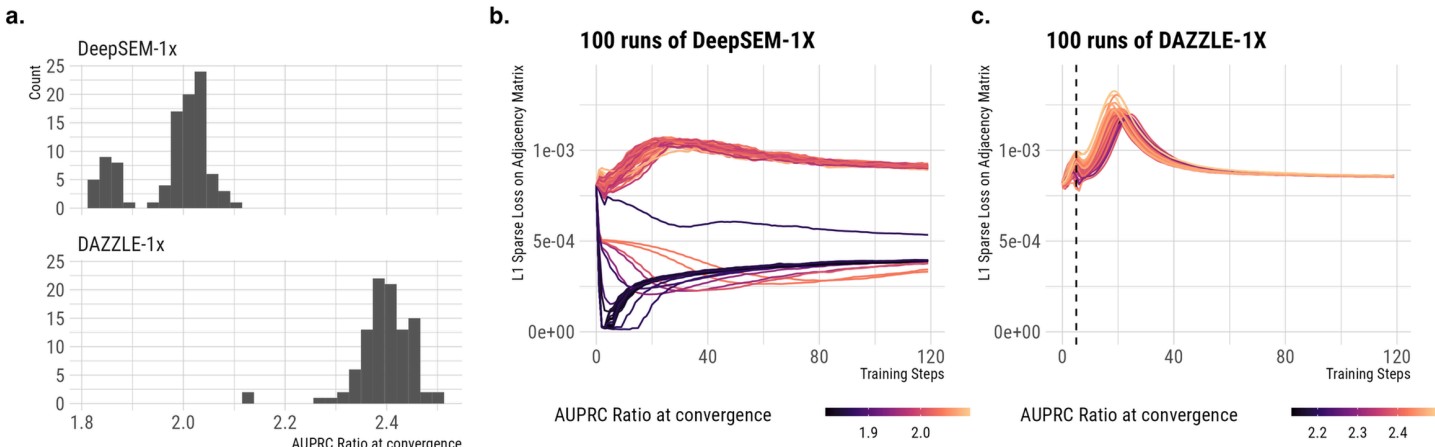

**Fig 3. Comparison of 100 runs of DeepSEM-1x and DAZZLE-1x on hESC evaluated using the STRING network.** In DeepSEM-1x, the prioritization of the L1 sparsity control at early stage is the main cause of unstable GRN inference performance. DAZZLE solves this issue by delaying the introduction of the L1 sparse loss by 5 iterations. **a)** Histogram of AUPRC ratios. **b)** Sparse loss over time for DeepSEM-1X, colored by AUPRR at convergence. **c)** Sparse loss over time for DAZZLE-1x, colored by AUPRR at convergence.

input genes. All edges whose weights have an absolute value over 0.001 are extracted and analyzed.

For this case study, we focused on validation using literature evidence, since our goal was to confirm those context-specific regulatory interactions. General-purpose regulation databases, such as TRRUST v2, are invaluable for general quantitative benchmarking (as described in Sect 3.3.1 and Table 1), but literature curation is better suited for verifying if the discovered links are relevant to the specific cell types and conditions analyzed here.

These GRN inference results confirm that gene regulation is a dynamic process that changes across the lifespan. Fig 4a lists the top ten regulated genes at each time point, ranked by the summed edge weights of regulating relationships on each gene. At the earliest life stages, most of the top regulated genes are associated with cell proliferation and differentiation. For example, *Tuba1a* (Tubulin alpha 1a) encodes proteins in microtubules, which form the mitotic spindle for cell division and motility structures that move cells to their correct positions. At later ages, however, we see more regulation of immune response genes. Many of these genes, such as *Tmem176B* (transmembrane protein 176B), *H2.D1* (histocompatibility 2, D region locus 1), and *PISD* (phosphatidylserine decarboxylase), encode key proteins, receptors, and enzymes of microglial immune response.

Fig 4b shows a closer view of two specific genes of interest. First, since *Tmem119* (transmembrane protein 119) is often used as a biomarker to differentiate microglia from other immune cells in the brain [36], we chose it as the center of the local network to analyze. Fig 4b shows that regulation of and by *Tmem119* is only heavily active starting in the late juvenile stage. Seven of the top 10 predicted regulators, *C1qc*, *Cd81*, *Cx3cr1*, *Hexb*, *Lgmn*, *P2ry12*, and *Selplg*, are commonly considered to be part of the microglial transcriptional "signature," as they are generally expressed at low levels in other immune cells [37–41]. In addition, *Csf1r* has recently been identified as a regulator of pathogenesis for microglia and macrophages [42]. It is reasonable to hypothesize that this surrounding local neighborhood includes much of the core functionality of healthy microglia.

*Apoe* (apolipoprotein E) is another well-studied gene that encodes a protein playing a central role in lipid metabolism, neurobiology, and neurodegenerative diseases. Unlike *Tmem119*,

## a. Top regulated genes at different mouse developmental stages

| Embryonic Day (E14) | Early Postnatal Day (P4/5) | Late Juvenile Stage (P30) | Adulthood (P100) | Old Age (P540) |
|---|---|---|---|---|
| Tuba1a | Rsrp1 | Serinc3 | Serinc3 | Lyz2 |
| Malat1 | Lpcat2 | Tgfbr1 | Rsrp1 | Hmha1 |
| P2ry12 | P2ry12 | Srrm2 | Tgfbr1 | PISD |
| H3f3b | Ltc4s | Actb | H2.D1 | Ly6e |
| Tubb5 | Serinc3 | Rbm39 | Frmd4a | Rbm39 |
| Apoe | Glul | Apoe | Rbm39 | Mbnl1 |
| Ywhae | H3f3b | Frmd4e | Ssh2 | H2.D1 |
| Fth1 | Asah1 | Mbnl1 | Ctsh | Sepp1 |
| C1qc | Crybb1 | Adap2 | Cd14 | Gpr34 |
| Trap1a | Malat1 | Rrbp1 | Tmem176b | Eef1a1 |

## b. Closer look at local networks around specific genes at different developmental stages

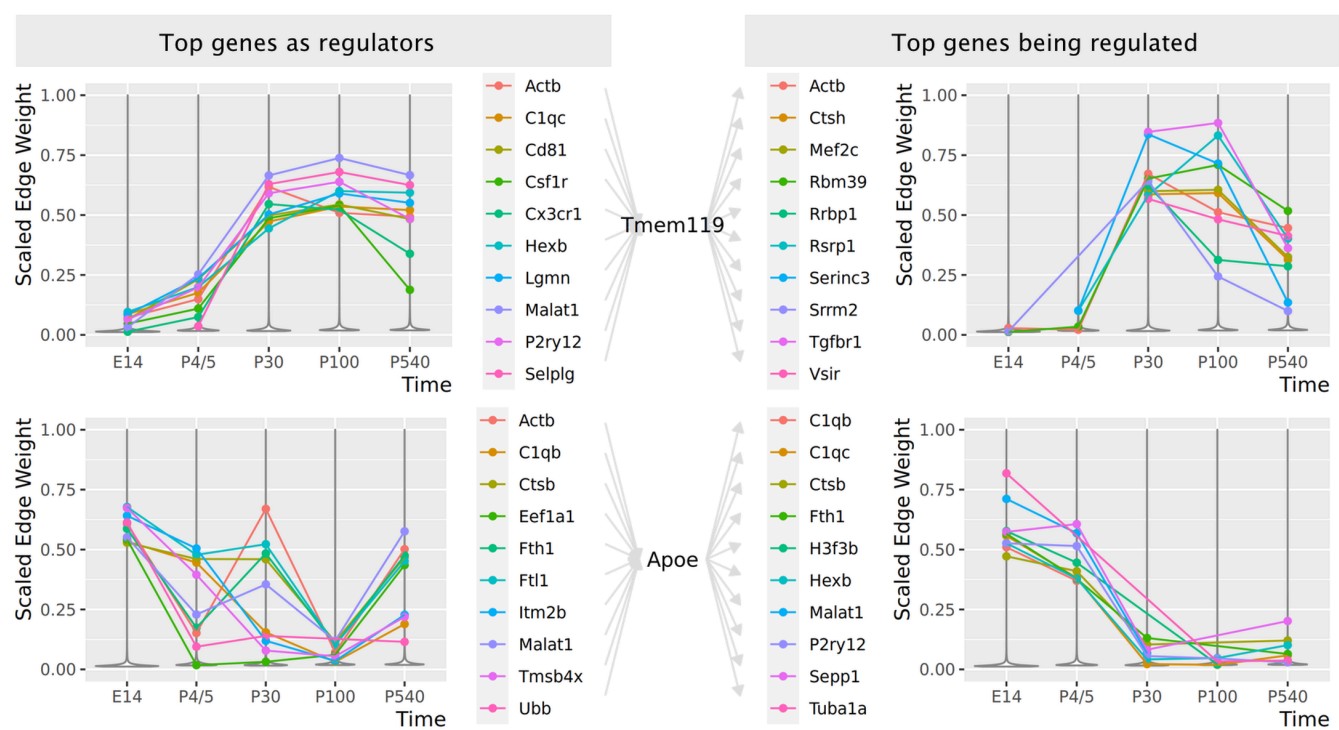

**Fig 4. a. Top 10 regulated genes at each life stage in mouse microglia.** b. Predicted local networks around *Tmem119* and *Apoe* in mouse microglia. Here edge weights are min-max scaled at each time point. Top genes are selected according to the maximum weight at all time points.

*Apoe* is highly regulated at embryonic day E14. As the mice mature, the relative impact of *Apoe* drops, but it increases again in old age. The list of top predicted links by DAZZLE is consistent with recent studies showing the variety of *Apoe*'s roles in cellular activities. For example, at E14, the top three genes regulating *Apoe* are *Ftl1* (ferritin light polypeptide 1), *Tmsb4x* (thymosin beta 4 X-linked), and *Itm2b* (integral membrane protein 2B). Although the connection between *Apoe* and *Ftl1* is not well established, a recent study [43] shows that *Apoe*

deficiency leads to increased iron levels. Another study [44] suggests iron loading is a prominent marker of activated microglia in Alzheimer's disease patients. Further, both genes are located on mouse chromosome 7, about 20cM apart. This evidence suggests a plausible regulatory connection between *Ftl1* and *Apoe* and further reflects the important role of iron in early brain development.

Fig 4 illustrates DAZZLE's predicted regulation patterns for these two typical well expressed genes. However, we have found that DAZZLE's regulatory predictions make sense even for genes whose overall expression levels are low. (See, e.g., comparable images for *Ifit3* (Interferon Induced Protein With Tetratricopeptide Repeats 3) on the project web site https://bcb.cs.tufts.edu/DAZZLE/hammond.html.)

Another molecule worth mentioning here is *Malat1* (Metastasis Associated Lung Adenocarcinoma Transcript 1), which appears in our predicted networks as one of the top regulators for many microglia core genes, including (Fig 4) *Tmem119*, *Apoe*, and *H2.D1*. As a long non-coding RNA (lncRNA), *Malat1* has been identified in many pathological processes with immunological components, including several types of cancer [45] and diabetes [46]. It has also been identified as a key regulator in the microglial inflammatory response [47,48], but beyond that its function in microglia is mostly unexplored.

Overall, our analysis of this data set confirms that DAZZLE can handle typical real-world single-cell data with more than 10,000 genes and thousands of cells. On examination, the predicted networks appear generally consistent with current research on gene regulation in microglia. Beyond previously identified links, DAZZLE also suggests novel yet plausible links that may be confirmed through future experiments.

## 3. Materials and methods

### 3.1. Dropout augmentation

Previous research suggests that zero values in single cell data include both biologically real zeros, corresponding to truly absent genes, and random dropout events. A successful single-cell model should remain robust regardless of how dropout values are distributed. This idea informs the dropout augmentation algorithm.

Let $X \in \mathbb{R}^{n \times m}$ be a gene expression matrix from a single-cell experiment, where $n$ is the number of cells and $m$ is the number of genes. We randomly sample a proportion ($p$) of data and temporarily replace these values with zeros. Alternatively, the augmented data could be treated as the sum of the original expression data $X$ and a dropout noise term $E$, where $E$ is the Hadamard product of negative X and a masking term derived from Bernoulli sampling. By denoting the mask of dropout augmentation as $M_{DA}$ and the probability of augmented dropout by $p$, we can write the dropout noise term $E$ and the augmented data $X'$ as follows:

$$E = -X \odot M_{DA}, \text{where } M_{DA}^{ij} \sim \text{Bernoulli}(p) \tag{1}$$

$$X' = X + E. \tag{2}$$

In theory, the dropout augmentation algorithm could be applied to any iterative learning algorithm, such as expectation maximization, gradient boosting, or neural networks. As shown in Fig 1, $M_{DA}$ is re-sampled at every training step, so the augmented data $X'$ changes in every iteration. During the entire training process, the model is only exposed to the altered $X'$ instead of $X$.

### 3.2. GRN inference with DAZZLE

The task of GRN inference is to infer a weighted adjacency matrix $A \in \mathbb{R}^{m \times m}$ based on the expression data $X$. Previous methods DAG-GNN [30] and DeepSEM [18] rely on a linear additive assumption that can be written as

$$X = XA + Z, \tag{3}$$

where $Z \in \mathbb{R}^{n \times m}$ is a random variable characterizing the noise, essentially describing the gap between the overall expected counts of the genes based on their regularizers ($XA$) and the observed counts ($X$). To embrace the idea that the observed data are noisy due to dropout, we modify this assumption and rewrite Eq 3 as:

$$X' = X'A + Z', \tag{4}$$

where $Z'$ is defined for $X'$ analogously to the definition of $Z$ for $X$. Since dropout is so prevalent in single-cell data, we believe Eq 4 describes the actual situation more accurately. Following a similar transformation to that done in DAG-GNN and DeepSEM, by rearranging the terms we can rewrite Eq 4 in the following two forms:

$$Z' = X'(I - A), \tag{5}$$
$$X' = Z'(I - A)^{-1}. \tag{6}$$

Eq 5 infers $Z'$ from $X'$ and Eq 6 is a generative model that reconstructs $X'$ based on the noise sum. These two equations naturally fit into a VAE framework with Eq 5 as an encoder and Eq 6 as a decoder. When we parameterize both the VAE model and the adjacency matrix, the encoder can be denoted by $q_\phi(Z'|X')$ and the decoder can be denoted by $p_\theta(X'|Z')$, with $A$ being part of $\phi$ and $\theta$. In this case, $Z'$ is the latent variable.

For a VAE, the problem of finding the set of parameters $\theta$ that maximizes the log evidence $\log(P(X'))$ is intractable. Instead, it is common to maximize the evidence lower bound (ELBO), which we write as

$$\begin{aligned} ELBO = &-D_{\mathrm{KL}}(q_\phi(Z'|X')\|p_\theta(Z')) \\ &+ \mathbb{E}_{z' \sim q_\phi(Z'|X')}[\log p_\theta(X'|Z')], \end{aligned} \tag{7}$$

where the first term is the KL divergence and the second term can be thought of as the reconstruction loss.

The random variable $Z'$ describes the deviation of the observed value $X'$ from the expected value $X'A$. In a particular cell, if the expression value of a particular gene happens to be observed as 0 due to dropout events, we will more likely see a larger deviation $Z'$. In other words, $Z'$ contains information that can be used to infer whether a value comes from a dropout event. Following this rationale, we can add a classifier $C_{\mathrm{DA}}$ based on the specified dropout augmentation masked $M_{\mathrm{DA}}$, as shown in Eq 8 below. As a naïve approach, here we choose a simple 3-layer multi-layer perceptron (MLP, activated by tanh) followed by sigmoid as the DA classification:

$$\hat{M}_{\mathrm{DA}} = \mathrm{sigmoid}(C_{\mathrm{DA}}(Z')). \tag{8}$$

The loss function of this classifier is simply a binary cross entropy function.

$$L_{BCE} = -\mathbb{E}\left[M_{DA}\log\hat{M}_{DA} + (I - M_{DA})\log(I - \hat{M}_{DA})\right] \tag{9}$$

The dropout augmentation classifier can be trained either separately or together with the main model using the same optimizer. In our experiments, we add the classification loss to the ELBO function scaled by a hyper parameter $\chi$. Additionally, following the design of DeepSEM, we include an L1 sparse loss term that regulates the sparsity of the learned adjacency matrix. The final form of the objective function is to minimize the following loss function:

$$
\begin{aligned}
\text{Loss} = &-\mathbb{E}_{z\sim q_\phi(Z|X)}\left[\log p_\theta(X|Z)\right] \\
&+ \alpha\sum|A| \\
&+ \beta D_{KL}(q_\phi(Z|X)||p_\theta(Z)) \\
&+ \gamma L_{BCE}(M_{DA}, \hat{M}_{DA}).
\end{aligned}
\tag{10}
$$

### 3.3. Datasets used

**3.3.1. BEELINE single-cell benchmarks.**   The BEELINE benchmarks consist of both synthetic expression data based on curated ground truth networks, as well as seven pre-processed real single-cell RNA-seq datasets [6]. These scRNA-seq datasets come from both human and mouse samples and have undergone different pre-processing steps, including normalization, depending on the original data format (e.g. raw counts or processed data). In some aspects, this variety reflects the wide array of differences we encounter in real-world data.

Next, BEELINE combines the scRNA-seq data with three different sources of "ground truth" data about regulatory relationships, including the functional interaction network represented in the STRING database v11 [49], non cell-type specific transcriptional networks, and cell-type specific networks. The Non-specific ChIPSeq network combines links from DoRothEA [50], RegNetwork [51], and TRRUST v2 [52]. The cell-type specific networks were created by the BEELINE authors for each dataset by searching through the ENCODE [53], ChIP-Atlas [54], and ESCAPE [55] databases. To generate a benchmarking dataset, BEELINE identifies highly variable transcription factors and genes and randomly samples from this pool to create a benchmark of the desired size.

**3.3.2. Hammond microglial data.**   To assess DAZZLE's performance in a more practical context, we use a published data set from [35] (data available from NCBI's Gene Expression Omnibus database [56] under accession number GSE121654). The Hammond mouse microglial dataset includes RNA sequencing counts for cells underlying several possible comparisons. In our analysis, we selected the data from five mouse developmental stages, each of which includes single cell data from four healthy male mice. To preprocess the data, following suggestions from [57] for the same data, we filtered out cells with fewer than 400 or more than 3,000 unique genes, cells with more than 10,000 UMIs, and cells with over 3% of reads mapping to mitochondrial genes. Note that here we adopt a cross-sectional slicing strategy and treated each developmental time point as independent samples. The top regulations for key regulators are then compared to existing literature to validate the biological plausibility of the inferred networks.

Many standard analysis approaches further reduce the data set size by filtering the gene set, keeping only the most variable genes. However, here, we only remove genes with a raw count of zero transcripts detected in all cells. We further removed all gene models, mitochondrial

genes, and ribosomal genes from this pool to simplify the interpretation of the resulting networks. The expression values were normalized using natural log transformation.

After this cell and gene filtering, the final data set includes 49,972 cells from five time points across the mouse lifespan: Embryonic (embryonic day E14, 15,673 genes and 11,262 cells), Early Postnatal (postnatal day P4/5, 13,316 cells and 15,039 genes), Late Juvenile (P30, 9,431 cells and 13,929 genes), Adulthood (P100, 8,259 cells and 13,998 genes), and Old Age (P540, 7,704 cells and 14,140 genes). Note that compared to the original paper [35], here we are using a very different approach, analyzing changes in potential regulatory links across time, rather than attempting to identify microglial subpopulations defined by specific injury-responsive cell clusters.

### 3.4. Evaluation metrics

In this report, we follow the recommendations from the BEELINE paper and use Area Under the Precision Recall Curve Ratio (AUPRC Ratio) and Early Precision Ratio as the main evaluation metrics. The main reason why AUPRC is preferred over Area Under the Receiver Operating Characteristic (AUROC) is that ultimately we are classifying a potential link between a TF and a target gene to be either exist or not exist. In the ground truth data, usually there are far more non-existing edges than existing edges. For example, in the hESC dataset, there are 5,149 edges in a data set with 578,100 potential edges. AUPRC is generally considered a better metric when there is a class imbalance between the positive and negative groups [58], as in this case. For easy comparison with other methods, we still provide results evaluated by the AUROC metric in Table C in supplement S1 Text. The AUROC results show similar trends to those of the other metrics.

In this paper, AUPRC is approximated using its discretized form without interpolation, as shown in Eq 11:

$$\text{AUPRC} = \int_0^1 P(R)\,dR \approx \sum_{n=1}^{N} P_n \cdot (R_n - R_{n-1}), \tag{11}$$

where $P(R)$ is the Precision at Recall level R, $N$ is the number of unique predictions, and $R_n$ and $P_n$ are the Recall and Precision scores at item $n$. This approximation is often referred to as average precision [59] and is the same analysis performed in the DeepSEM paper. The AUPRC Ratio is calculated by dividing the calculated AUPRC score with the theoretical AUPRC score of a random predictor. In this case, the expected precision of a random predictor is equal to the proportion of positive cases. On the Precision-Recall plot, the performance of a random predictor forms a horizontal straight line (see Fig 5D in [58]). Therefore, the AUPRC of a random predictor is simply the proportional of positive cases and we can calculate AUPRC Ratio with Eq 12.

$$\text{AUPRC Ratio} \approx \frac{\text{Positives}}{\text{Total Instances}} \cdot \sum_{n=1}^{N} P_n \cdot (R_n - R_{n-1}) \tag{12}$$

Early Precision (EP) is the fraction of true positives within the top-k candidates, where k is the number of edges in the ground truth network. Early Precision Ratio further divides that fraction with the expected Early Precision of a random predictor, which is the edge density of the ground truth network [6].

Since the number of all possible edges and the number of true edges are usually very large in actual GRNs, the values of the AUPRC and Early Precision themselves tend to be small. As

claimed in the BEELINE paper, converting them to ratios make it easier to understand the performance across benchmarks.

## 4. Discussion

In this study, we tackle the dropout problem in real-world single-cell data paradoxically by adding more dropout. As previously mentioned, while the idea of using dropout to improve the robustness of machine learning models has existed for a long time, thanks to the pioneering work of Bishop and Hinton [28,29], it has rarely been discussed and recognized in the 'omics community as a useful method although our data suffers greatly from sparsity and noise. Our work shows that it can be useful in the task of GRN inference, but it seems to be a reasonable assumption that it would also help many other applications in the single-cell domain with robustness and perhaps improved performance at very low cost. Of course, as in other machine learning domains, dropout augmentation is not a magic bullet that can guarantee a significant amount of performance gain. But it is clear that that Dropout Augmentation can improve model robustness, which is equally important, making observed performance gains believable. Building on the principles established in this paper, we have since developed RegDiffusion [31], a follow-up method that frames the noise augmentation concept within a more formal diffusion model framework. Instead of adding dropout noise, RegDiffusion incrementally adds Gaussian noise and learns to reverse the process. This subsequent approach offers further improvements in computational speed and inference accuracy, demonstrating the extensibility of the core idea introduced in DAZZLE.

In the specific case of GRN inference, our proposed method DAZZLE not only stabilizes the output but also produces better predictions. On the BEELINE benchmarks, a single-run of DAZZLE yields comparable results to an ensemble containing of 10 repeated trials of the previously most-accurate method. Our experiment with the microglia dataset shows that DAZZLE has the capacity to run on large single-cell data with minimal gene filtration. While we do not have ground-truth networks suitable for direct comparison in these cases, the predicted networks are consistent with current understanding and include plausible novel links. These novel links may be good candidates for future investigations of key regulatory relationships.

Finally, DAZZLE, like GENIE3 and GRNBOOST2, only requires the gene expression matrix as the input. Therefore, since DAZZLE has better performance and runs faster, it can be seemlessly used in popular downstream GRN analysis tools such as SCENIC [13,60] and SCENIC Plus [61]. As suggested in the SCENIC papers, the results of DAZZLE could be pruned using cisTarget data [62] to remove unlikely edges and further improve the accuracy. The learned regulation could be used to calculate AUCell scores[13] to describe the TF activities for each cell and to build dimension reduction plots for cells. We provide a tutorial on how to integrate the results from DAZZLE and RegDiffusion into the SCENIC workflow on the documentation site for RegDiffusion (https://tuftsbcb.github.io/RegDiffusion). We are also developing a GPU-based SCENIC calculation pipeline called flashscenic (https://github.com/haozhu233/flashscenic).

One limitation of this method is that it can not directly predict if the regulation is positive or negative. One of the advantages of DeepSEM and DAZZLE is that they use neural networks to do nonlinear transformation. At the end, the learned adjacency matrix is in fact describing the relationships among the non-linearly transformed data. Therefore, positive/negative regulation is not directly computable. One possible solution to this problem would be, after the existence of regulation is confirmed, to create another simple linear model between the

TF and the target and use the sign of the linear model as the direction of the inferred relationship.

A practical limitation of the current model architecture is that the space complexity of this model also scales quadratically. With 15,000 genes, the model requires 30 Gb GPU memory, which still fits in a single modern GPU. However, for even larger use cases, the method may require multiple GPUs. Another limitation is that the current version is designed to be applied to each individual dataset (time point or cell cluster). Thus, it does not help us develop a universal understanding of gene interactions. How to relax these restrictions, how to learn these connections in a more efficient way, and how to interpret the inferred networks biologically are important questions to consider in future work.

## Supporting information

**S1 Fig. Additional ablation study shows the usefulness of Dropout Augmentation.** Certain proportions of data points (x-axis) were drop to simulate background dropout noise at the very beginning.
(TIFF)

**S2 Fig. Quality of inferred GRN from DeepSEM-1x: AUPRC Ratio as a function of the number of training iterations.** Quality may quickly downgrade after the performance peak. Dashed line is the recommended stopping point from DeepSEM. Note that for the celltype-specific data sets, performance is particularly volatile.
(TIFF)

**S3 Fig. Distribution of 100 runs of DAZZLE-1x and DeepSEM-1x on BEELINE evaluated using the STRING network.** Results from DAZZLE-1x tend to be more stable than results from DeepSEM-1x.
(TIFF)

**S1 Text. Addition supplement information including hyperparameter choices and additional benchmarks.**
(PDF)

## Acknowledgments

We thank Liping Liu, Rebecca Batorsky, Yijie Wang, and Teresa Przytycka for their thoughtful comments. We also appreciate the help of Hantao Shu, Jianyang Zeng, and Jianzhu Ma for sharing their data and code from the original DeepSEM paper.

## Author contributions

**Conceptualization:** Hao Zhu, Donna K. Slonim.

**Data curation:** Hao Zhu.

**Investigation:** Hao Zhu.

**Methodology:** Hao Zhu.

**Project administration:** Donna K. Slonim.

**Resources:** Hao Zhu.

**Software:** Hao Zhu.

**Supervision:** Donna K. Slonim.

**Validation:** Hao Zhu, Donna K. Slonim.

**Visualization:** Hao Zhu.

**Writing – original draft:** Hao Zhu, Donna K. Slonim.

**Writing – review & editing:** Hao Zhu, Donna K. Slonim.

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
