## [Decision Letter · Decision Letter 0]

4 Feb 2025

PCOMPBIOL-D-24-01735

Improved gene regulatory network inference from single cell data with dropout augmentation

PLOS Computational Biology

Dear Dr. Zhu,

Thank you for submitting your manuscript to PLOS Computational Biology. As with all papers, your manuscript was reviewed by members of the editorial board. Based on our assessment, we have decided that the work does not meet our criteria for publication and will therefore be rejected. If external reviews were secured, reviewers' comments will be included at the bottom of this email.

We are sorry that we cannot be more positive on this occasion. We very much appreciate your wish to present your work in one of PLOS's Open Access publications. Thank you for your support, and we hope that you will consider PLOS Computational Biology for other submissions in the future.

Yours sincerely,

Saurabh Sinha

Academic Editor

PLOS Computational Biology

Jian Ma

Section Editor

PLOS Computational Biology

**Additional Editor Comments (if provided):**

The reviewers have raised substantial concerns about the computational innovation in the work, as well as empirical demonstration of the new method's advantage over established methods.

**Reviewers' Comments (if peer reviewed):**

Reviewer's Responses to Questions

**Comments to the Authors:**

Reviewer #1: In this manuscript, the authors considers inferring gene regulatory networks (GRNs) from single cell data. The authors introduce a strategy, DA, that dropouts some random data points. Then DA is applied to enhance a known GRN inference method, DeepSEM, and the new method is named DAZZLE. DAZZLE is tested on some data sets, and it performs better than other methods on BEELINE data sets.

DAZZLE performs better than DeepSEM. Why can you claim that the improvement is from solving the zero inflation problem? For deep learning models, interpretability is always an issue, and it is difficult to state that the improvement is due to certain reasons. One reasonable solution is: for a good data set without many zeros, apply DeepSEM to obtain performance measurement 1. Then randomly dropout some data points and apply DAZZLE to obtain performance measurement 2. If measurement 2 is close to or better than measurement 1, then you can claim that the zero inflation problem is solved.

Section 2.5, DAZZLE is applied to data at each stage of development. If I understand correctly (cf. Eq.3), DeepSEM and DAZZLE work for gene expression data at stationary. However, gene expression during development is not very stationary. In this case, are the temporal changes of GRN real, or is DAZZLE applied to an improper situation and produces unreliable results?

DAZZLE chooses to delay the introduction of sparsity loss, and stops training after 120 iterations. Since the training procedure is not described in detail, I am a little confused: did you do hyperparameter fine-tuning (including the dropout augmentation probability) and evaluation on the same data sets? If so, how can you guarantee that the good performance is not due to overfitting?

The code is well-organized. Good job.

Besides the above problems, the novelty (minor modifications of a published method) might not meet the standard of PLOS CB. I think it is better to revise it thoroughly and submit to a lower-level journal.

In introduction, explain in detail about the data type. What type of data is used for training? What types of data can DAZZLE be applied to? For data types, you just specify that they should be scRNAseq. Are they measured at one time point, or multiple time points (or you need to infer pseudo-time)? If you need data from multiple time points, do you need to measure the same cell multiple times? Do you need the cells to be intervened (i.e., driven away from stationary) before measurement?

In introduction, explain whether your method can determine the direction of regulation between gene i and gene j. Also, does it distinguish between activation and inhibition?

The first time that you use “adjacency matrix”, mention that it is the GRN you want to infer.

The introduction should contain a little more details about your methods: roughly describe the structure of your model, and the training data. Readers who haven’t read the DeepSEM paper might find that the introduction is not informative enough, and the whole paper is difficult to read.

L.112, DAZZLE looks self-contained. Raise an example about what other network components can be integrated with DAZZLE.

BEELINE paper recommends GENIE3, GRNBoost2, and PIDC. Any reason that PIDC is not compared?

L.125-126: grammar problem.

AUROC and AUPRC are both commonly used in the evaluation of GRN inference methods. Any reason that AUROC is not reported? (L.386-390 explains that AUPRC is better than AUROC, although I personally prefer adding AUROC, since it has some good properties.)

Explain more about BEELINE. What is STRING?

L.292-293, each position has an independent probability p to be dropped? Rewrite this sentence in a rigorous way (as you did in L.296-299).

L.301, training step: is this the same as epoch?

Eq.4: how can you prevent A=I? Add some explanations.

Eq.6: why is I-A invertible?

Eq.9, as BCE, why is it I (identity matrix) - M, not ones(m, n) - M?

What if a position is 0 by natural dropout, and it is reset to 0 by the mask?

Eq.11: n is already used as cell number. Use another notation. Also, it seems that the accurate expression is \sum (P_n + P_{n-1}) / 2 \times (R_n - R_{n-1}). Is your approximation necessary?

About the measurements, there is another issue: can your method distinguish between positive regulation (activation) and negative regulation (inhibition)? Do the ground truth GRNs in your datasets distinguish them? If yes, AUPRC needs to be modified since it is designed for binary classification.

Reviewer #2: In the manuscript, the authors proposed the innovative Dropout Augmentation (DA) method and the DAZZLE model to address the zero-inflation issue in single-cell RNA sequencing data, enhancing the accuracy and stability of gene regulatory network (GRN) inference. The DA strategy regularizes data by adding simulated dropout noise, and the DAZZLE model, under the structural equation model framework, uses the variational autoencoder architecture with optimizations like adjacency matrix control, structure simplification and closed-form priors. However, several points need further discussion:

1. Single-cell multi-omics data for GRN inference is a leading trend. Despite using the BEELINE dataset for benchmarking scRNA-seq-based GRN inference methods, the authors should explore the DAZZLE model’s feasibility and effectiveness in leveraging multi-omics data and its integration for better GRN inference.

2. Figure 1 shows DAZZLE’s advantage over DeepSEM due to DA. Authors should clarify the DA classifier’s role and impact on the final GRN inference results to help readers understand its contribution.

3. It would be beneficial for the authors to discuss DAZZLE’s unique advantages over existing methods, especially those brought by the Dropout Augmentation technique. Notably, its role in enhancing GRN inference stability warrants in-depth exploration, such as analyzing how it reduces inference errors and perturbations for more reliable GRN predictions.

4. Table 1 indicates DAZZLE’s performance boost over other methods, yet explanations focus on dropout and structural enhancements, lacking in-depth analysis of how they translate to accurate GRN predictions. Theoretical and empirical justifications for this are needed.

5. As seen in Table 1, DAZZLE was compared with classic methods. Given the rapid methodological advances in GRN inference, further comparison with cutting-edge methods is recommended to showcase DAZZLE’s innovation.

6. The authors used AUPRC and AUPRC Ratio to evaluate DAZZLE, a common practice. But considering the complexity of GRN inference, more comprehensive metrics like AUPR, F1-Score and AUROC should be incorporated to fully evaluate and present the capabilities and limitations of DAZZLE.

Reviewer #3: The main innovation of this manuscript lies in the proposal of the Dropout Augmentation (DA) method, which can effectively address the zero-inflation issue in scRNA-seq data. In addition, this manuscript combines the DA with the DeepSEM method to propose the DAZZLE model. Both DeepSEM and DAZZLE are gene regulatory network inference methods based on structure equation model. The differences between the above two methods are the use of DA. My primary concerns about this manuscript are the innovativeness and effectiveness, which is detailed as follows.

1. Since the main innovation of this manuscript is DA, it would better demonstrate the importance of DA if the authors could combine DA with other methods instead of just with DeepSEM. Another reason for this consideration is that the current innovation of this paper is relatively modest. The biggest difference compared to DeepSEM is DA, while DeepSEM is a study published in 2021, which is relatively old in terms of publication year.

2. I have observed that this manuscript, following the recommendation from BEELINE, has adopted AUPRC and EPR for performance demonstration. However, BEELINE also utilized AUROC. I recommend that the authors present the experimental results for AUROC. Although AUPRC is more appropriate for assessing imbalanced datasets than AUROC, it is worth noting that AUROC, being the most common evaluation metric in the field of GRN inference, can reflect the predictive ability of the model.

3. According to the results in Table 1, DAZZLE did not achieve the best performance on some datasets. For instance, when inferring the STRING network, GENIE3 exhibited the optimal performance on the hHEP and mDC datasets; when inferring non-specific networks, GENIE3 showed the best performance on the mESC and mHSC-GM datasets; and when reconstructing cell type-specific networks, GENIE3 achieved the best performance on the hHEP dataset. These experimental results indicate that the predictive performance of DAZZLE cannot surpass that of GENIE3, which was published in 2010.

4. The comparison methods are GENIE3, GRNBoost2, and DeepSEM. The authors could include more studies published after 2021 for comparison.

**Have the authors made all data and (if applicable) computational code underlying the findings in their manuscript fully available?**

Reviewer #1: Yes

Reviewer #2: None

Reviewer #3: None

PLOS authors have the option to publish the peer review history of their article (what does this mean?). If published, this will include your full peer review and any attached files.

Reviewer #1: No

Reviewer #2: No

Reviewer #3: No

---

## [Decision Letter · Decision Letter 1]

10 Aug 2025

PCOMPBIOL-D-24-01735R1

Improved gene regulatory network inference from single cell data with dropout augmentation

PLOS Computational Biology

Dear Dr. Zhu,

Thank you for submitting your manuscript to PLOS Computational Biology.

Two of the original referees re-reviewed your revision: one of them approves (Reviewer 2) and the other (Reviewer 1) was not convinced. Reviewer 3 did not respond to the re-review request, so we obtained an additional opinion from a fourth reviewer who is favorable, with some minor suggestions. Given the mixed reviewer opinions and our own reading of your revision, we invite you to submit a revised manuscript addressing the following minor comments. Your responses will be verified by the editorial team, for a quicker turnaround time.

Reviewer 1: “Explain that each entry of the gene expression matrix represents the scRNAseq result, where the original value x is replaced by log(1+x). Otherwise, readers might wonder how 0 is handled.”

Reviewer 1: “PIDC method has a GitHub page: https://github.com/Tchanders/network_inference_tutorials. What do you mean by “this package is no longer available”?”

Reviewer 4: Comments 1, 2, 6.

(Please see reviewer comments below.)

Please submit your revised manuscript within 30 days Oct 10 2025 11:59PM. If you will need more time than this to complete your revisions, please reply to this message or contact the journal office at ploscompbiol@plos.org. Please include the following items when submitting your revised manuscript:

We look forward to receiving your revised manuscript.

Kind regards,

Saurabh Sinha

Academic Editor

PLOS Computational Biology

Jian Ma

Section Editor

PLOS Computational Biology

**Journal Requirements:**

1) We note that your Figures and Supplementary Figures files are duplicated on your submission. Please remove any unnecessary or old files from your revision, and make sure that only those relevant to the current version of the manuscript are included.

2) Your manuscript's sections are not in the correct order.  Please amend to the following order: Abstract, Introduction, Results, Discussion, and Methods.

3) We notice that your supplementary figures are uploaded with the file type 'Figure'. Please amend the file type to 'Supporting Information'. Please ensure that each Supporting Information file has a legend listed in the manuscript after the references list.

**Reviewers' comments:**

Reviewer's Responses to Questions

Reviewer #1: Some comments in my previous round of review (especially those regarding math) were neither discussed in the rebuttal letter nor followed in the manuscript. Why?

The advantage of deep learning methods in GRN inference is still questionable. Besides, the math of DeepSEM is not quite reasonable. Therefore, DA as a simple improvement of DeepSEM (or other supervised learning methods) might not be sufficiently significant. I think DA does not really solve the zero inflation problem (see below), which also decreases its significance.

My suggestion is to revise it and submit it to a lower-level journal. Also, the authors should respect every review comment.

DA and zero inflation: Figure S1 shows that with or without zero inflation, DA can improve DeepSEM. However, DA cannot fully cancel the effect of zero inflation. The authors should lower the tone. Instead of the false impression that “DA can solve the zero inflation problem”, it is better to state that DA can improve performance even with zero inflation.

Development is impossible if the dynamics of gene expression (not GRN) is at stationary. The application of DAZZLE in Section 2.5 is not satisfactory, especially because there is only qualitative verification, not quantitative evaluation.

Explain that each entry of the gene expression matrix represents the scRNAseq result, where the original value x is replaced by log(1+x). Otherwise, readers might wonder how 0 is handled.

PIDC method has a GitHub page: https://github.com/Tchanders/network_inference_tutorials. What do you mean by “this package is no longer available”?

Reviewer #2: The authors have addressed my concerns and I have no further comments.

Reviewer #4: The manuscript introduces DAZZLE, a method that incorporates Dropout Augmentation to reconstruct gene regulatory networks using scRNA-seq data from human or mouse. This approach refines the existing DeepSEM framework into a new pipeline and benchmarks its performance against several ground truth GRNs from the popular BEELINE toolkit. The study presents a comprehensive evaluation using both ground truth and real-world datasets. The following minor comments will further strengthen the manuscript.

[Major comments]

1. The authors claimed improved efficiency and reduced computational time for DAZZLE. Line 108: “These changes lead to reduced model sizes and computational time”. And Lines 168-169: “a single pass of DeepSEM or DAZZLE can take several hours to compute, even on modern GPUs”. While it may not be feasible to provide detailed computational time or GPU specs, the authors could strengthen their claim by including example run times or comparative results to illustrate the efficiency gains.

2. “For datasets with a large number of genes and cells, a single pass of DeepSEM or DAZZLE can take several hours to compute, even on modern GPUs. In such cases, a single pass of DAZZLE offers a sufficiently-good solution at a more reasonable cost.” It is unclear whether the first mention of DAZZLE refers to DAZZLE-10x and the second to DAZZLE-1x. If so, please clarify, as the current phrasing appears contradictory.

3. In Section 2.5, in addition to literature evidence, the authors may consider databases such as TRRUST v2 and TFLink to query experimentally validated TF-gene interactions. This would allow for a more quantitative evaluation of DAZZLE’s performance by reporting how many ground truth TF-gene interactions were identified.

[Minor comments]

4. DeepSEM and DEEPSEM were used interchangeably in the manuscript.

5. Line 137, “celltype-specific ChiP-Seq” should be “celltype-specific ChIP-Seq”.

6. Lines 454-460, references are needed for this paragraph. For example, cisTarget and AUCell scores haven’t been mentioned in the manuscript.

**Have the authors made all data and (if applicable) computational code underlying the findings in their manuscript fully available?**

Reviewer #1: Yes

Reviewer #2: None

Reviewer #4: Yes

PLOS authors have the option to publish the peer review history of their article (what does this mean?). If published, this will include your full peer review and any attached files.

Reviewer #1: No

Reviewer #2: No

Reviewer #4: No

**Figure resubmission:**
---

## [Editor Report · Decision Letter 2]

9 Oct 2025

Dear Zhu,

We are pleased to inform you that your manuscript 'Improved gene regulatory network inference from single cell data with dropout augmentation' has been provisionally accepted for publication in PLOS Computational Biology.

Best regards,

Ferhat Ay, Ph.D

Section Editor

PLOS Computational Biology

---

## [Editor Report · Acceptance letter]

PCOMPBIOL-D-24-01735R2

Improved gene regulatory network inference from single cell data with dropout augmentation

Dear Dr Zhu,

I am pleased to inform you that your manuscript has been formally accepted for publication in PLOS Computational Biology. Your manuscript is now with our production department and you will be notified of the publication date in due course.

With kind regards,

Anita Estes
